# Lp(a) and the Risk for Cardiovascular Disease: Focus on the Lp(a) Paradox in Diabetes Mellitus

**DOI:** 10.3390/ijms23073584

**Published:** 2022-03-25

**Authors:** Karam M. Kostner, Gerhard M. Kostner

**Affiliations:** 1Department of Medicine, University of Queensland, Mater Hospital, Brisbane 4102, Australia; k.kostner@uq.edu.au; 2Institute of Molecular Biology and Biochemistry, Medical University of Graz, A-8010 Graz, Austria

**Keywords:** lipoprotein(a), diabetes mellitus, type-1 DM, type-2 DM, metabolism, transcriptional regulation, atherosclerosis, cardiovascular disease, thrombosis, medication

## Abstract

Lipoprotein(a) (Lp(a)) is one of the strongest causal risk factors of atherosclerotic disease. It is rich in cholesteryl ester and composed of apolipoprotein B and apo(a). Plasma Lp(a) levels are determined by apo(a) transcriptional activity driven by a direct repeat (DR) response element in the apo(a) promoter under the control of (HNF)4α Farnesoid-X receptor (FXR) ligands play a key role in the downregulation of *APOA* expression. In vitro studies on the catabolism of Lp(a) have revealed that Lp(a) binds to several specific lipoprotein receptors; however, their in vivo role remains elusive. There are more than 1000 publications on the role of diabetes mellitus (DM) in Lp(a) metabolism; however, the data is often inconsistent and confusing. In patients suffering from Type-I diabetes mellitus (T1DM), provided they are metabolically well-controlled, Lp(a) plasma concentrations are directly comparable to healthy individuals. In contrast, there exists a paradox in T2DM patients, as many of these patients have reduced Lp(a) levels; however, they are still at an increased cardiovascular risk. The Lp(a) lowering mechanism observed in T2DM patients is most probably caused by mutations in the mature-onset diabetes of the young (MODY) gene and possibly other polymorphisms in key transcription factors of the apolipoprotein (a) gene (APOA).

## 1. Introduction

Indisputably, cholesterol-rich lipoproteins are amongst the most significant risk factors for atherosclerotic disease, including coronary heart disease, myocardial infarction, stroke, and peripheral vascular diseases. The cholesterol-ester-rich lipoproteins are comprised mostly of low-density lipoproteins (LDL) and lipoprotein(a) [Lp(a)]. As will be discussed in more detail later, Lp(a) consists of an LDL core particle and the specific antigen, apolipoprotein(a) [apo(a)]. Despite the considerable structural homology between Lp(a) and LDL, the genetics, metabolism, and pathophysiology of these two lipoproteins are very different [1,2]). While a great deal of information is available concerning the pathophysiology of atherosclerotic diseases, many gaps remain in our knowledge. One such question is why two different individuals with the same set of risk factors can show different incidences and courses of this disease. On the other hand, there is no doubt that additional risk factors can significantly increase the risk of cardiovascular disease (CVD). This is particularly so for the major risk factors; elevated LDL-cholesterol (LDL-C), Lp(a), type-2 DM mellitus (T2DM), hypertension, smoking, and chronic inflammation. As an example, we have postulated some 40 years ago that patients with Type-II hyperlipoproteinemia (grossly elevated LDL-C) plus a serum Lp(a) concentration exceeding 30 mg/dL are at a >10-fold risk for myocardial infarction, in comparison to individuals with Lp(a) > 30 mg/dL and normal LDL-C.

Many original papers, reviews, and books have been published on various aspects of Lp(a) research, each reflecting certain time periods. Additionally, there are numerous papers indicating that patients with high Lp(a) plus T2DM are particularly at an increased risk for atherosclerosis [3]. In contrast, however, publications exist demonstrating that T2DM patients have lower Lp(a) plasma levels in comparison to individuals without T2DM. We call this phenomenon “The Lp(a) Paradox in Diabetes Mellitus,” and we will attempt to identify the possible causes for these observations in the current review.

## 2. Historical Background

In the first publications by the group of Kare-Berg in 1963, Lp(a) was called “sinking-pre-β-lipoprotein (SPB)” [4]. At the time, lipoproteins were separated by electrophoresis or ultracentrifugation, and Berg observed an extra pre-β1 band in lipoprotein electrophoresis that, in contrast to conventional LDL, was found in the high-density lipoprotein (HDL) fraction by density gradient ultracentrifugation (Figure 1).

The features of SPB were unknown, yet there were several polymorphisms known of apoB-100 [5], the major protein of LDL, and SPB was thought to be one of them. A comprehensive review of the history of Lp(a) has been published more recently [1], and in the next few paragraphs, we highlight certain names that will remain unforgettable in Lp(a) research. Kare Berg, Gösta Dahlen, and Martin Frick first described Lp(a) as a serum factor found in individuals with CHD. [6]. Angelo Scanu, Gunther Flerss, and Celinda Edelstein published on the structure of Lp(a) and demonstrated that its protein moiety consists of one molecule of apoB-100 and a specific glycoprotein, apo(a), linked together by a disulfide bridge [7]. Scanu’s research group was also the first to realize that apo(a) is partially homologous to plasminogen. In fact, when Richard Lawn and John McLean cloned apo(a) in 1987 [8], it became evident that apo(a) consists of numerous repetitive entities, so-called “kringles,” that show up to 90% homology to the corresponding kringles in plasminogen. This discovery opened avenues for molecular biologists and geneticists. Gerd Utermann, together with his collaborators Hans Dieplinger and Hans-Jörg Kraft, was the first to realize that apo(a) exhibits a previously unknown size heterogeneity in proteins [9].

Utermann et al. [9] provided evidence that size heterogeneity is caused by the presence of a variable number of kringle-IV repeats in individual Lp(a) samples. They also identified numerous polymorphisms and mutations within the coding region and the promoter region of the *APOA* gene. These mutations turned out to be responsible for the great variation of Lp(a) concentrations found in different individuals, as well as in various ethnic groups [10].

The homology of apo(a) and plasminogen inspired the labs working in fibrinolysis and blood clotting: The laboratories of John Chapman and Angles-Cano found that Lp(a) interferes with the activation of plasminogen to plasmin by t-PA and, in turn, its binding to fibrin clots [11]. Other prominent names in this area were Edward Plow, John Gaubatz, Robert Hegele, Joseph Loscalzo, and Joel Morriset. A burning question at that time was the mechanism of Lp(a) biosynthesis, assembly, and catabolism. First investigations along these lines were published by Ann White and Robert Lanford [12], who addressed these questions in liver cell cultures of baboons and demonstrated that apo(a) is biosynthesized independently of LDL: During secretion, apo(a) binds to the surface of liver cells and assembles with low-density lipoproteins in the space of Disse.

The interference of Lp(a) with hemostasis and fibrinolysis was, in part, the basis for addressing its role in atherosclerotic disease. In fact, our research group was amongst the first to show that individuals with Lp(a) plasma concentration >30 mg/dL are at a significantly increased risk for myocardial infarction, and this risk rises exponentially, not only with plasma concentration of Lp(a) but also with that of LDL [13]. Although there is a cut-off concentration (30 mg/dL) at which Lp(a) is considered to be an independent and causal risk factor for CVD, there is currently great uncertainty in the analytical procedures for Lp(a) that are routinely used in clinical laboratories. The first methods were elaborated by John Albers and Santica Marcovina [14] using radio-immunoassays or ELISA’s. Since these methods are not entirely suitable for automated high-throughput screening, a working group was formed, headed by Christa Cobbaert and Liesbeth Deprez, to elaborate reference methods and reference material for standardizing and harmonizing Lp(a) assays (https://www.ifcc.org/ifcc-scientific-division/sd-working-groups/wg-apo-ms/, accessed on 4 February 2022). This standardization of Lp(a) assays will obliterate the numerous discrepancies that exist in results published by individual research groups.

## 3. Lp(a) Structure

In different types of electrophoresis, Lp(a) is displayed as a lipid stainable, distinct band in the pre-β1 region between β- and pre-β (LDL and VLDL) lipoproteins. With ultracentrifugation, most Lp(a) is found in the HDL_1_ region, although, depending on the isoform, Lp(a) may also hide in the LDL or HDL2 fractions. Additionally, in the plasma of heterozygote individuals with two distinct polymorphic apo(a) forms, two Lp(a) bands are frequently observed by density gradient ultracentrifugation. It is important to note that lipoproteins are purified mainly from the fasting plasma of healthy individuals. Under these conditions, an idealized Lp(a) particle is mostly found to consist of an LDL core particle surrounded by one apo(a) glycoprotein (Figure 2A). Similar pictures can be found in numerous review articles on Lp(a).

In normolipemic fasting plasma, approximately 75% of the immune reactivity is found in the HDL_1_ region, while the rest distributes amongst VLDL, LDL, whole HDL, and the bottom fraction. In non-fasting plasma, the distribution of apo(a) is even more heterogeneous. This is particularly true in plasma with elevated triglyceride-rich lipoproteins (TGRLp), as observed in patients with T2DM. The exact structural features of Lp(a) found outside the HDL fraction have not been fully explored. However, unpublished work from our laboratory suggests that TGRLp contains apo(a) that is not fully assembled with apoB-100, in addition to some Lp(a) aggregates. We have also revealed that apo(a)-containing fractions isolated by immune-adsorbers consist partly of fractions with apoB:apo(a) ratios >1 or, in other words, LDL:Lp(a) complexes. Although the apo(a) found in the lipid-free bottom fraction after ultracentrifugation may contain small amounts of full-length apo(a), most of it consists of apo(a) fragments created by proteolytic enzymes [15].

Extensively purified Lp(a), isolated from the fasting plasma of healthy donors by several consecutive purification steps, consists of one LDL core particle (that is indistinguishable from LDL of density 1.016–1.063) and one apo(a) glycoprotein, linked together by one disulfide bridge. The composition of Lp(a) in comparison to LDL is shown in Table 1 below.

Apo(a), the specific antigen of Lp(a), has a very characteristic structure and shares close homology to plasminogen (Plg) [8,11]. In addition to a protease domain, Plg has five distinct kringle sections, numbered I-V (Figure 3). Apo(a) cDNA is 75–100% homologous to Plg, with a variable number of kringle-IV (K-IV), one K-V-like domain, and a protease-like domain. The numerous K-IVs in apo(a) are only partly identical; in fact, ten different subtypes of K-IVs have been found. Where K-IV-1 and K-IV-3—K-IV-10 (the so-called “unique kringles”) are present only once, a variable number of K-IV-2 exist amongst different individuals. This is the characteristic feature responsible for the size heterogeneity of apo(a) observed in different individuals, where up to 50 K-IVs have been identified.

There is an urgent dispute in the literature regarding the units of measurement for Lp(a). In most publications up to approximately 2015, Lp(a) was expressed in mg/dL, and a cut-off between 30–40 mg/dL was assumed for CVD. Given that the composition of Lp(a) is extremely variable, it was concluded that concentration units should only be expressed in nmol/L. While this is indisputably correct, most high throughput Lp(a) assays are based on immune turbidimetric or nephelometric methods using polyclonal antibodies. Due to the variable number of K-IV repeats, such methods overestimate the concentration of large isoforms and underestimate that of small isoforms. Therefore, it is not a straightforward method to apply one conversion factor of mg/dL into nmol/L for plasma samples with Lp(a) of different isoforms. An additional problem may be that apo(a) is a glycoprotein with a carbohydrate content of approximately 28%, and this is mostly neglected when calculating the true molecular mass.

The theoretical molecular mass of apo(a) with 20 K-IV repeats, including the carbohydrate moiety, is 368,016.26 Daltons (D). For each additional K-IV, 20,361 D must be applied. The molecular mass of the core LDL is variable, yet an average value of 2.8 million D is propagated in the literature. Thus, the mass of Lp(a) with 21 K-IV repeats is roughly 3.17 million D. On the basis of this value, a theoretical conversion factor of 3.15 (1 mg/dL = 3.15 nmol/L) may be calculated.

Recently, the theoretical mass of Lp(a) with 6 to 35 K-IV-2 repeats was calculated by Cobbaert and Ruhaak [17] to range from 2821 to 3344 kD. At the basis of these values, the calculated conversion factors are 3.54–2.99. It must be stated here that manufacturers of Lp(a) assays propagate much lower conversion factors between 2.2–2.4. Therefore, it is evident that further research in this area is crucial in order to determine fixed widely acceptable Lp(a) units.

## 4. Lp(a) Metabolism

### 4.1. Biosynthesis and Assembly

Apo(a) is biosynthesized almost exclusively in the liver, yet small amounts of apo(a) mRNA have also been identified in the brain and testis [8]. The significance of these two latter organs in Lp(a) metabolism remains obscure.

In early investigations, we studied the turnover of Lp(a) in healthy individuals and demonstrated for the first time that the plasma Lp(a) concentration highly and significantly correlates with the rate of biosynthesis. However, no correlation could be found with the fractional catabolic rate (FCR) [18] (Figure 4).

In comparison, in individuals with elevated LDL-C, the FCR of Lp(a) correlated significantly with that of LDL (Figure 5).

Although the reason for this observation was not fully explored, it is tempting to assume that the rate of LDL biosynthesis might be responsible for these observations. Furthermore, the plasma Lp(a) concentration may also be driven by the speed of LDL production. Our observations have been confirmed in numerous subsequent studies and are most relevant in strategies for pharmacological interventions in patients with hyper-Lp(a): Drugs for lowering plasma Lp(a) must reduce apo(a) biosynthesis and/or the Lp(a) assembly, whereas naturally increasing its catabolism will most likely fail.

The expression of the *APOA* gene follows general principles of transcription → translation → post-translational modifications and secretion from cells. For gene transcription, positive and negative regulatory elements are key. We addressed this question and identified >70 response elements in the apo(a) promoter for transcription factors and nuclear receptors. The significance of apo(a) expression in most of them is still unknown, yet the most important one could be characterized by promoter expression studies [16]. The basis of these studies was the observation that patients with obstructive jaundice and elevated plasma levels of bile salts exhibit grossly reduced plasma Lp(a) levels. Bile acids are cognate ligands for the Farnesoid-X receptor (FXR). In a series of experiments, we came up with the following picture: FXR ligands have a dual impact on apo(a) gene expression driven by the canonical transcription factor HNF4α. We identified several putative binding sites for hepatic nuclear factors (HNFs) in the apo(a) promoter, yet the region −826 to −814 relative to the transcription initiation was most prominent. The binding of ligands to FXR causes its translocation to the nucleus, whereby HNF4α binding to its response elements is inhibited, followed by a cessation of apo(a) expression. In a second, equally important pathway, FXR promotes the expression of fibroblast growth factor-19 (FGF-19) in the intestine. FGF-19 is transported to the liver then binds to its cognate receptor, FGF-19 R4. This activates a signaling cascade involving a MAP kinase cascade RAS-ERK1/2 and binding of phosphorylated ELK-1 to an ETS promoter segment located at −1603 to −1615. Together with pathway-1, FXR activation leads to almost complete inhibition of apo(a) transcription. The pathways described above are schematically displayed in Figure 6.

After transcription and translation, apo(a) is heavily –*N* and –*O* glycosylated and passes the Golgi apparatus ready for secretion. As mentioned above, under normal conditions, >95% of apo(a) found in plasma is bound to genuine LDL that is sparsely found in the liver but rather derives from TG hydrolysis of VLDL in circulating blood. Thus, the question arises where and how the assembly of native Lp(a) occurs. Early research from our group revealed that a genuine Lp(a) might be synthesized in vitro in the test tube by mixing purified LDL with recombinant apo(a) in the absence of any cofactor. It was, therefore, speculated that in vivo, apo(a) gets secreted from the liver and binds in a similar way to apo(a) by the interference of lysine groups on apoB-100 with specific kringle domains in apo(a). For this first step, K-IV-3 and K-IV-6 appear to be most relevant. In a second step, the apo(a):apoB-100 complex is stabilized by a disulfide bridge formed between Cys-3000 in apoB and the only free lysine group in K-IV-9. Early work published by White and Lanford [19], however, only partially supported this hypothesis, as they demonstrated by using baboon liver cell cultures, that apo(a) during secretion is bound to the cell surface. Upon contact with mature LDL, these two proteins associate and form a genuine Lp(a).

However, this extracellular model of assembly has been challenged by stable isotope turnover studies in vivo. The group of Hans Dieplinger infused 3-fold deuterated lysine D_3_-Lys into humans and followed its differential incorporation into apo(a) and apoB-100 over time [20]. The results of these experiments demonstrated that the enrichment of D_3_-Lys in the apoB-100 of LDL was significantly faster than that of apoB-100 in apo(a), yet the D_3_-Lys incorporation velocity into apo(a) and apoB-100 from Lp(a) was identical. These findings are mostly compatible with an intracellular assembly of Lp(a). Overall, it must be admitted that both mentioned pathways displayed in the diagram in Figure 7 are appealing, yet none of them have so far been unambiguously confirmed.

### 4.2. Lp(a) Catabolism

After the publication of the pioneering work by Brown and Goldstein [21] that indicated LDL is catabolized by a specific receptor-mediated pathway, a significant number of distinct lipoprotein receptors, aside from the LDL-R, have been published. These comprise the “metabolic receptors,” such as LDL-R, VLDL-R, chylomicron remnant receptor, apoE receptor, and more. In addition, it became apparent that modified and oxidized lipoproteins are cleared by different scavenger receptors, such as SR-A, SR-B1, Lox-1, and more [1] (Figure 8). With regard to the catabolic pathway of Lp(a), there are reports that all these receptors bind Lp(a) in vitro in cell cultures, yet their significance for the validity of these findings in vivo presently remains unclear.

Since one of the major proteins in Lp(a) is apoB-100, we studied the binding, internalization, and degradation of Lp(a) in cultured human fibroblasts and found that there was a definitive binding and degradation that could be competed off by LDL [22]. In in vivo turnover studies, on the other hand, we followed the decay of radio-labeled Lp(a) in comparison to LDL over time in normo-lipemic individuals and in one homozygous patient suffering from familial hypercholesterolemia (FH) who was deficient of LDL-receptors. While the residence time of Lp(a) in plasma of normo-lipemic individuals was almost twice as long as that of LDL, the fractional catabolic rates of Lp(a) and LDL were identical in the FH patient. These results are strong indications that the LDL-R pathway is not operative in vivo for Lp(a).

The question now arises as to how Lp(a) might be catabolized. Since it is not ethical to investigate lipoprotein uptake by different organs in humans in vivo, we performed such studies of the uptake of radiolabeled human Lp(a) into different organs of laboratory animals, including mice, rats, rabbits, and hedgehogs, and found that the majority of Lp(a), approximately 50–60%, winds up in parenchymal liver cells [23]. The remainder was found in the bile, spleen, and kidney. This led us to speculate that many of the receptors mentioned above, to some extent, play a role in Lp(a) removal from circulation. More recently, two receptors that are not specific for lipoproteins, namely the asialoglycoprotein receptor (ASGP-R) [24] and the plasminogen receptor, both highly abundant on liver cells, turned out as strong candidates for their role in Lp(a) catabolism.

The ASGP-R is responsible for removing “aged” glycoproteins from circulation that might have been modified after a long residence time. In fact, many glycoproteins possess sialic acid as terminal sugar and, after its cleavage by neuraminidases, the penultimate sugar mannose-amines of galactose-amine get exposed and are strongly bound by the ASGP-R on liver cells and removed. This might also occur with Lp(a), as we were able to demonstrate that even native Lp(a) is bound to some extent to ASGP-R positive, but not by ASGP-R negative fibroblasts. These findings have also been verified by in vivo studies in rats. After treatment of Lp(a) with neuraminidase in vitro and injected into rats, we observed a very fast uptake and catabolism by the liver (Figure 9).

The second receptor of note is the Plg receptor, PlgRKT. The group of McCormick published an interesting work in 2017 in Circulation Research [25], providing strong evidence that the PlgRKT present on liver cells binds a great deal of Lp(a). This is not surprising since apo(a) is highly homologues to Plg. The most interesting results of these studies, however, were that Lp(a), after binding and internalization into lysosomes, dissociates into LDL, which is degraded. The liberated apo(a) migrates from Rab5+ early endosomes to the trans-golgi network and Rab11+ recycling endosomes and finally is secreted in an intact, un-degraded form. The recycled apo(a) probably re-assembles outside the liver with apoB-100, forming a new Lp(a). Since it is known that recycling proteins, such as transferrin and apoE, play physiological roles in transporting ligands into cells of specific organs, the authors speculated that this in-fact might be the function of apo(a), namely, to shuffle substances such as oxidized phospholipids or fibrin fragments into corresponding organs. The results of these experiments are highly relevant for interpreting the data of in vivo metabolic studies. It would mean that there exist two pools, one consisting of newly biosynthesized apo(a) and the other of recycled apo(a), and both apo(a) pools must have striking different metabolic parameters. It will be challenging to dissect these two pathways in future research and clarify their role in the overall metabolism under normal conditions and under the influence of different medications.

## 5. Lp(a)—One of the Most Atherogenic Lipoproteins

In 1981, we published findings that individuals with Lp(a) levels exceeding 30–50 mg/mL are at a significantly elevated risk for myocardial infarction [13]. This risk rises exponentially with the Lp(a) concentration and the abundance of LDL or other cognate risk factors for cardiovascular diseases. In the last 40 years, there has been a discrepancy concerning the significance of Lp(a) as a CVD-risk factor, and some of the reports were rather controversial. This was mainly caused by problems in methodology for Lp(a) measurements. Today, there is no doubt that elevated Lp(a) is causally related to MI and stroke, and this is documented by a great number of prospective epidemiological studies. It is unfortunately beyond the scope of this review to list all the eminent work that led to this recognition. Therefore, we may mention here only the most convincing results of the Copenhagen laboratories, published by Tybjaerg—Hansen, Langsted, Kamstrup, and Nordestgaard [26,27]. In prospective studies lasting for >10 years with collectives comprising >100,000 individuals from Denmark, the Copenhagen group provided unequivocal proof of the following facts:Lp(a) is an independent causal risk factor for cardiovascular disease, myocardial infarction, and stroke. This was unequivocally proven by Mendelian randomization assessment of the results.Lp(a) is equally atherogenic in male and female individuals. Multivariable-adjusted hazard ratios (HRs) for MI for increasing Lp(a) levels were 1.1–3.6 in women and 1.5–3.7 in men.The risk for CVD and MI rises with increasing Lp(a) concentrations with hazard ratios (HR) of 1.2 with Lp(a) levels between the 22nd and the 66th percentile and an HR of 2.6 for individuals >95th percentile.Lp(a) is a strong risk factor for aortic valve calcification.Based on these results, the European Atherosclerosis Society recommended in a consensus paper a cut-off level for Lp(a) of 50 mg/dL, which is approximately the 80th percentile in the European population.

### 5.1. Pathophysiological Aspects

The question of why Lp(a) might be the strongest risk factor for CVD and even more significant than LDL, has only partly been answered. Amongst all the available theories, we mention here only those that we consider the most plausible ones:Affinity to cell surface components: Early studies provided evidence that Lp(a) binds to glycosaminoglycans (GAG) in the arterial intima with a 4-fold higher affinity compared to LDL. Thus, arteriosclerotic plaques are full of Lp(a):GAG complexes that attract immune-competent cells, notably monocytes and macrophages, and trigger inflammatory processes [28].Interference with clotting factors and fibrinolysis, reviewed in [29]. Due to the high number of kringles present in apo(a), Lp(a) has a high affinity to Lys groups of fibrin. Fibrin: apo(a) complexes, as they are found in atherosclerotic plaques, have been shown to be partly resistant to lysis by plasmin.Apo(a) interferes with the activation of plasminogen by t-PA in the formation of plasmin (11).Lipid oxidation: There is no doubt that oxidative stress and free radicals are hallmarks of atherogenesis. There are numerous pathways described in that respect; however, the one that is relevant for the atherogenicity of Lp(a) is caused by free radicals oxidizing phospholipids on cell surfaces. Ox-PL is avidly bound and taken up by Lp(a). Such modified Lp(a), known as Lp(a)-Ox, accumulates through GAG binding in blood vessels and leads to inflammatory processes, as well as cytokine and interleukine liberation and starts an immunological process that triggers plaque formation [30].

### 5.2. Factors That Have an Impact on Plasma Lp(a) Levels

Unlike other lipoproteins, Lp(a) concentrations in plasma are only marginally influenced by diet. Drugs affecting Lp(a) concentrations will be treated in a later chapter, but it may be stated here that conventional drugs such as statins or fibrates are mostly ineffective. A more detailed treatment of this topic is found in Ref. [16]. Since Lp(a) is synthesized in the liver, it is of no surprise that liver diseases have a profound influence on Lp(a) production. The strongest effect is found in patients with intra- or extra-hepatic cholestasis and elevated bile acids in the blood that may be caused by gallstones or cancer. In these patients, the Lp(a) levels may go down to almost zero, yet they return back to “normal” if the patients are successfully treated.

In addition to the liver, the kidney is also a key organ in Lp(a) metabolism. A summary of this topic is found in Ref. [31]. Patients with chronic renal failure have up to 3-fold higher Lp(a) levels or more compared to healthy individuals. Glomerulonephritis is also accompanied by extremely high Lp(a). In chronic renal failure, the Lp(a) elevation might be caused by impaired catabolism, while patients with nephrotic syndrome exhibit an increased Lp(a) synthesis. 

The kidney also secretes rather large proteolytic fragments of apo(a) that consist mostly of the N-terminal part of apo(a) with K-IV Type-1 and -2. We have calculated that the amount of Lp(a) removed from circulation comprises nearly 1–2% of the total catabolism [32]. Since the amount of apo(a) fragments found in urine highly significantly correlates with the plasma Lp(a) concentration, we proposed that urine instead of serum might be taken for assessing the CVD risk [33]. This would likely solve many problems inherent to the current measurement procedures of Lp(a) in clinical laboratories. 

### 5.3. Pharmacotherapy of Elevated Lp(a)

There is a consensus of the European Atherosclerosis Society that Lp(a) levels >50 mg/dL (corresponding to approximately 120 nmol/L) represent a risk factor for CHD and require treatment. In a recent review article, Eraikhuemen et al. [34] comprehensively addressed this topic. For a long time, no medication was available that substantially lowered Lp(a). The most effective cholesterol-lowering drugs, such as statins, increase Lp(a) concentration in many patients. One exception, namely nicotinic acid, lowered Lp(a) up to 35% if applied in rather high dosages (3–5 g/day). However, due to the known side effects of this drug, nicotinic acid is off the market in many countries.

The proprotein convertase subtilisin/kexin type 9 (PCSK9) inhibitors, a relatively new class of cholesterol-lowering drugs, act by inhibiting the intracellular proteolytic degradation of the LDL-receptor and in turn promote >90% recycling. Consequently, LDL-C may be lowered by up to 50% and more. Interestingly, PCSK-9 inhibitors also lower plasma Lp(a) by 10–30%, yet the mechanism of their action on Lp(a) has not been fully explored [35].

Since a 30% reduction is considered insufficient for patients with very high Lp(a), the search for more efficient medications is ongoing. Fortunately, scientists have been successful in designing a drug based on antisense oligonucleotide (ASO) technology. Mipomersen, one of these classes of drugs, interferes with the biosynthesis of apoB and reduces LDL-C by up to 35% and apo(a) by some 25%. This drug is prescribed to patients resistant to statin or PCSK-9 treatment, yet its effect on Lp(a) is far from sufficient. More efficient antisense therapies have been developed that are specifically directed against apo(a) biosynthesis, for example, AKCEA-APO(a)-L_Rx_^®^, a second-generation ASO. Depending on the dose and frequency of application, AKCEA-APO(a)-L_Rx_ was capable of reducing Lp(a) levels between 50–80% without showing a rebound effect [36]. These drugs are currently in Phase-III clinical trials, and once they are approved for human treatment, they are due to provide final proof for the postulated causality of Lp(a) as a risk factor for MI, CVD, and stroke.

## 6. Lp(a) and Diabetes Mellitus (DM)

DM is a multifactorial and multigenetic disease and, as evidenced in the last decade, the characterization of patients with malfunctions of glucose (Glc) metabolism is far more complicated than originally thought. In the past, DM was mostly classified superficially and divided into two types: Type-1, characterized by the lack of insulin production, and Type-2, characterized by insulin resistance; alternatively, they were also frequently called juvenile diabetes mellitus and mature-onset diabetes mellitus. Today we know that there are numerous facets found in both types of DM that are either genetically determined, acquired, or both. A key element in DM is the glucose concentration in blood under fasting conditions and post-prandially. Simply speaking, the blood-glucose concentration is a result of its rate of biosynthesis and its rate of catabolism. The metabolic pathways of Glc biosynthesis and secretion into the blood are textbook knowledge and may not be reiterated here. Concerning its catabolism, there are many pathways that must be considered, ranging from uptake into cells of various organs involving glucose transporters (GLUTs), some of them being insulin-dependent, the burning of Glc for energy supply, the excretion of Glc by the kidney into urine and many more. One can imagine that in all the anabolic and catabolic pathways, a wealth of enzymes and their corresponding genes are involved that impact the pattern of DM. A key element in regulating blood glucose, without a doubt, is insulin. As shown in the diagram in Figure 10, insulin is produced in β-cells of pancreatic islets and stored in particular granula. When Glc concentration increases in blood, GLUT-2 is activated and promotes its influx into β-cells of the pancreas. In the Glc-sensing pathway, Glc is phosphorylated and follows the known glycolysis pathway, whereby the ATP concentration in islet cells increases. At high concentrations, ATP-sensitive potassium channels close and depolarize the β-cell. This promotes the influx of Ca++ and, in turn, the degranulation of the insulin-containing particles, followed by insulin release. In type-1 DM, dysfunction of any of the components in this pathway is conceivable. In most cases, however, type-1 DM is caused by autoimmune damage of the insulin-producing β-cells. The diagram in Figure 1 also highlights the numerous pathways where parallels with apo(a) expression are conceivable.

The pathomechanisms in T2DM are quite distinct from that of T1DM. The classical form of T2DM is characterized by hyperinsulinemia caused by the resistance of the relevant organs, muscles, and adipose tissue to take up Glc in response to sensing insulin. Based on contemporary genetic methods, close to 100 polymorphisms and mutations relevant for the etiology of T2DM have been identified, and this sheds some light on the complexity of this disease [37,38]. These features led C. Herder and M. Roden to recently propose a novel typology of DM [39]. They clustered the phenotypes of DM into five different diabetes subgroups, two relevant to T1DM and three to T2DM. Although this classification still represents an oversimplification concerning genotypes, they certainly will help to improve the differential diagnosis and treatment protocol for patients suffering from DM.

Regarding the role of DM in Lp(a) metabolism, there are only a few solid studies available that may allow us to pinpoint the basic mechanisms related to alterations of Lp(a) levels in patients with T1DM or T2DM, while there are no studies that reflect the phenotypic or genotypic heterogeneity. We are therefore left with the only solution to simplify matters and report on general features of Lp(a) in T1DM and T2DM. 

It is commonly accepted that amongst the major risk factors for atherosclerosis and CVD, T2DM is one of the most important. The pathophysiological causes here are manifold and comprise, among others, inflammatory and immunological processes, hypertension, oxidative stress, and the presence of atherogenic lipoproteins. Here, we focus on Lp(a), the most atherogenic lipoprotein. Although numerous reports and reviews have been published, they are only partly consistent, and many are controversial. The reader may be alerted to two of the most recent reviews that summarize the findings until now [3,40]. 

S. Haffner [41] summarized the literature up to 1992 and concluded that they might not be valid in all points today: (i) Patients with T1DM have mostly elevated plasma Lp(a) levels that are related to their metabolic control. (ii) Microalbuminuria is a hallmark for high plasma Lp(a). (iii) T2DM patients, well-treated or not, have Lp(a) levels that are not different from controls. iv) There was no evidence that Lp(a) might increase the CVD risk in DM. In the years following 1992, many reports dealt with similar questions, but the answers were only in part coherent. The reason for this is based on the following facts: the methodology for Lp(a) measurements was not standardized, the studies did not dissect patients with impaired kidney function (which is known to have a strong influence on plasma Lp(a) levels), and the metabolic control and type of treatment was not accounted for in many investigations. In the following chapters, we endeavour to provide a global picture of the current situation.

### 6.1. Lp(a) in T1DM 

There have been very few studies published on this topic within the last ten years. The questions to be answered here are whether T1DM patients have Lp(a) plasma concentrations different from control individuals and, if yes, what are the mechanistic explanations for that? Results of more recent studies in T1DM are not yet available, and taking all available literature into consideration, a final answer to these questions cannot presently be provided. We can only summarize from data reported in Refs. [34,35,36] and state that: (1) There is no inherent effect on plasma Lp(a) levels caused by T1DM. (2) T1DM patients that are well-controlled have comparable Lp(a) levels to controls. (3) T1DM patients suffering from microalbuminuria and, more strikingly, patients with kidney disease, have increased plasma Lp(a) levels and (4) physical activity and healthy lifestyle normalizes elevated plasma Lp(a) in T1DM patients who have normal kidney function.

### 6.2. Lp(a) in T2DM

The situation of Lp(a) in T2DM is far more complicated. Since both Lp(a) and T2DM are strong risk factors for atherosclerosis, one would expect that this might be reflected by elevated Lp(a) levels. However, publications consistently report lower plasma Lp(a) in T2DM patients compared to controls. We call this the Lp(a) paradox in type-2 diabetes mellitus. T2DM is frequently accompanied by hypertension, altered lipid metabolism, elevated VLDL, hyperuricemia, hyperinsulinemia, inflammation, oxidative stress, as well as genetic polymorphisms in Glc transporters, nuclear receptors and more. All these factors have been shown to influence the metabolism of Lp(a).

### 6.3. The Lp(a) Paradox in T2DM

In our first investigations in 1981, regarding the role of Lp(a) in myocardial infarction, we found that high Lp(a) is not only a risk factor in normo-lipemic individuals but also to a much greater extent in individuals with elevated LDL-C [13]. In contrast, in individuals with Type-IV hyperlipoproteinemia, where patients consistently show an impaired Glc tolerance or T2DM, Lp(a) appeared to be a “negative risk factor,” i.e., MI patients had lower Lp(a) levels than controls (Figure 11).

At the time, we had no plausible answer for this observation. In 2013, a Mendelian Randomization study by Kamstrup and Nordestgaard was published [42] where, in a collective of some 80,000 individuals, they measured the plasma concentration of Lp(a) and, in addition, the number of KIV-2 repeats and the rs10455872 single nuclear polymorphism (SNP), in order to answer the question of whether the low plasma Lp(a) levels in T2DM might be causal or not. T2DM patients had lower Lp(a) concentrations with an odds ratio of 1.26. Individuals with high numbers of KIV-2 repeats (that correlates with low plasma Lp(a) levels) showed a higher risk for T2DM. On the other hand, carriers of the rs10455872 SNP associated with elevated Lp(a) concentrations did not show a different risk of T2DM. The authors concluded that low Lp(a) concentrations by themselves might not be causal for increased T2DM risk, yet this might differ for individuals with a high number of KIV-2 repeats.

After 2013, several other groups reported on plasma Lp(a) concentrations in T2DM patients, and most of these studies found lower Lp(a) compared to control individuals [43]. The obvious question now is what causes the reduced Lp(a) levels in T2DM? As indicated above, the situation is complex due to the numerous factors that influence the phenotype of T2DM, many of them related to mutations or polymorphisms of genes involved in lipid and lipoprotein metabolism. A good example of this complex situation is found in the article of Shih et al. [44], who studied the Q268X mutation in the MODY gene in relation to plasma apoAII, apo CIII, and Lp(a) levels. MODY stands for mature-onset diabetes of the young, and MODY genes are nuclear receptors (HNF1α and HNF-4α), known as master regulators of genes expressed in the liver, and are involved in lipid metabolism. As previously mentioned in the paragraph “transcriptional regulation of apo(a),“ the expression of apo(a) is highly dependent on the binding of HNF4α to a DR-1 response element in the promoter. Thus, any mutation in HNF4α that affects the transactivation of genes must have an influence on plasma lipid and lipoprotein levels. In fact, it was found that carriers of the Q268X mutation not only suffer from MODY but also have reduced plasma concentrations of Lp(a), apoAII, and apoCIII. There are other mutations and polymorphisms known in the MODY genes that may have similar effects on plasma Lp(a). Of further relevance are the findings that T2DM patients show aberrations in hormones other than insulin, such as testosterone, IFG-1, or thyroid hormones, all of which are known to impact *APOA* expression [45].

In summary, it appears that T1DM patients have Lp(a) concentrations that are not different from healthy individuals if they are well-controlled and free of kidney dysfunction. T2DM patients, on the other hand, may have reduced Lp(a) due to mutations or polymorphisms in genes that affect the expression of the APOA gene on the one hand and the phenotype of DM on the other.

### 6.4. Lp(a) as a Risk Factor for CAD in Patients with DM

In theory, Lp(a) should be at least as atherogenic, if not more, in diabetic patients than in non-diabetics. Lp(a) contains large amounts of oxidized phospholipids, a hallmark of atherogenesis. Due to its longer residence time in the blood compared to LDL [46], Lp(a) is probably glycated to a larger extent than LDL, thus contributing to its atherogenicity. That this occurs in vivo is supported by the findings of Kotani et al. [47] who demonstrated impaired endothelial function likely related to oxidized Lp(a) from T2DM patients. The theoretical considerations mentioned above have also been corroborated in patient studies in vivo.

In 2006, Kollerits et al. [48] questioned to what extent Lp(a) might be an independent predictor of CVD in IDDM patients. More than 400 IDDM patients were followed over an observation period of 10.7 years. Since renal disease is a significant risk factor for CAD, patients with impairments of kidney function were excluded from the study. Although this study did not answer the question as to whether or not IDDM patients have increased Lp(a) levels, it was concluded that Lp(a) values >30 mg/dL contribute significantly to the CAD risk in T1DM. Similarly, calcified aortic valve disease was found to occur more frequently in T1DM patients with high Lp(a) [49].

There are numerous reports documenting that the situation in T2DM patients with respect to the atherogenicity of Lp(a) is very similar to that of T1DM. In one of them, Saeed et al. [50] examined the association of Lp(a) with the risk for CVD in close to 10,000 male and female participants, 1543 of them suffered from diabetes or pre-diabetes. From the results, the authors concluded that “Adding lipoprotein(a) to traditional risk factors improved ASCVD risk prediction.” Interestingly, in a recent study by Markus et al. [51], it was reported that the relative increase in mortality from CVD was significantly higher in women with T2DM compared to men with T2DM.

Concluding from studies published so far, it appears that elevated plasma Lp(a) levels in T2DM patients positively correlates with the incidence of atherosclerotic cardiovascular disease. Despite the Lp(a) paradox in T2DM, there is no indication that lowering Lp(a) might negatively affect the cardiovascular outcome of this disease.

## Figures and Tables

**Figure 1 ijms-23-03584-f001:**
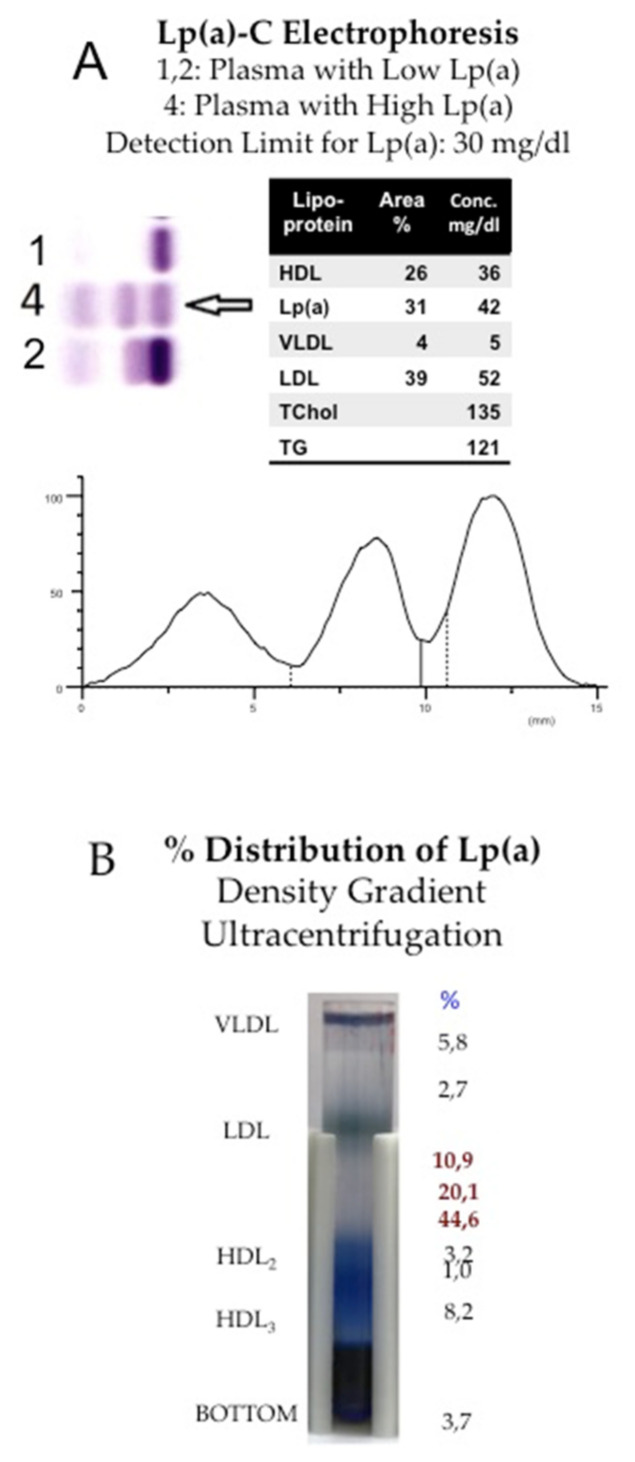
Separation of Lp(a) by electrophoresis or by density gradient ultracentrifugation. (**A**): Lp(a) migrates in gel electrophoresis as an extra pre-β1 band and can be quantitated either after staining for lipids with Sudan black or by staining with a cholesterol reagent. Here Lp(a) is separated in a routine laboratory by the Helena^®^ Electrophoretic system, and the concentration is given as Lp(a)-cholesterol. Given the fact that Lp(a) consists of some 25–30% of cholesterol, Lp(a) mass in mg/dL may be obtained by a factor of 3–4. (**B**): The heterogeneity of Lp(a) may be demonstrated by density gradient ultracentrifugation. The serum proteins were pre-stained with Coomassie blue, and lipoproteins were separated in the SW-41 Rotor, Beckmann^®^ for 24 h at 40,000 RpM. In this plasma sample, some 75% of Lp(a) was found in the HDL_1_ region, while the rest were distributed from the top to bottom fraction.

**Figure 2 ijms-23-03584-f002:**
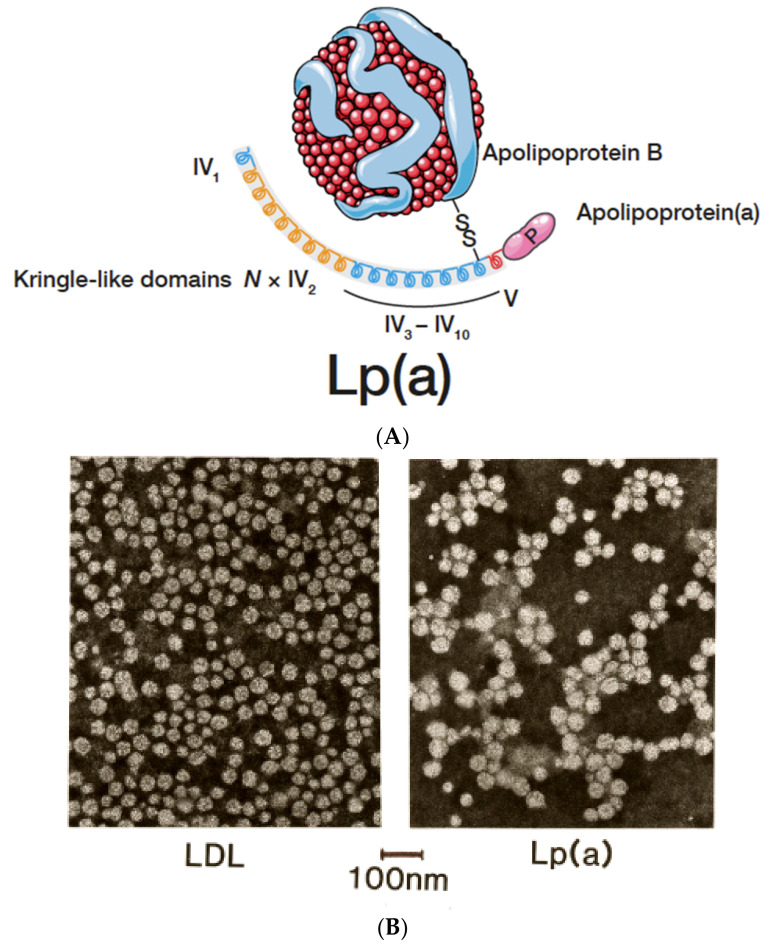
(**A**): Lp(a) consists of an LDL-core particle with apolipoprotein B-100 as the major protein, linked to glycoprotein apo(a) by a disulfide bridge. For more details, see text. (**B**): Negative stain electron microscopy of LDL and Lp(a). Although both particles look comparable, the diameter of Lp(a) is somewhat larger than LDL.

**Figure 3 ijms-23-03584-f003:**
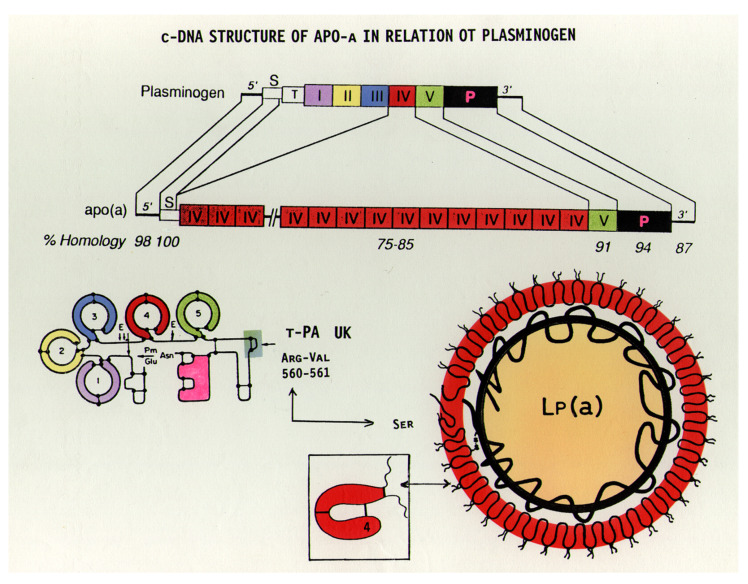
The cDNA structure of plasminogen in comparison to apo(a). While plasminogen has a single kringle-IV (K-IV) domain, a similar K-IV domain in apo(a) is repeated several times. Both structures possess a single K-V-like domain. In the protease domain of apo(a), the arginine of plasminogen is replaced by serine and therefore cannot be activated by t-PA or urokinase, as in the case of plasminogen [16].

**Figure 4 ijms-23-03584-f004:**
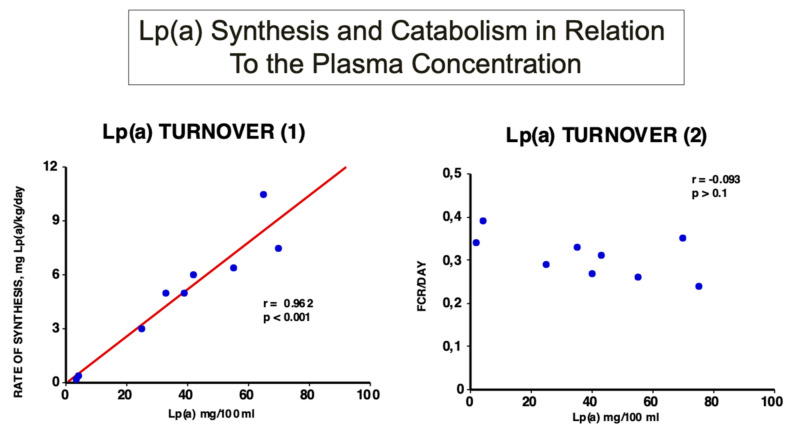
Turnover studies of Lp(a) in humans: Labelled and purified Lp(a) was injected into nine healthy probands with Lp(a) concentrations ranging from 5–84 mg/dL, and the decay over time was followed. As can be seen, the rate of Lp(a) biosynthesis, and not the fractional catabolic rate, correlates significantly with the plasma concentration [16].

**Figure 5 ijms-23-03584-f005:**
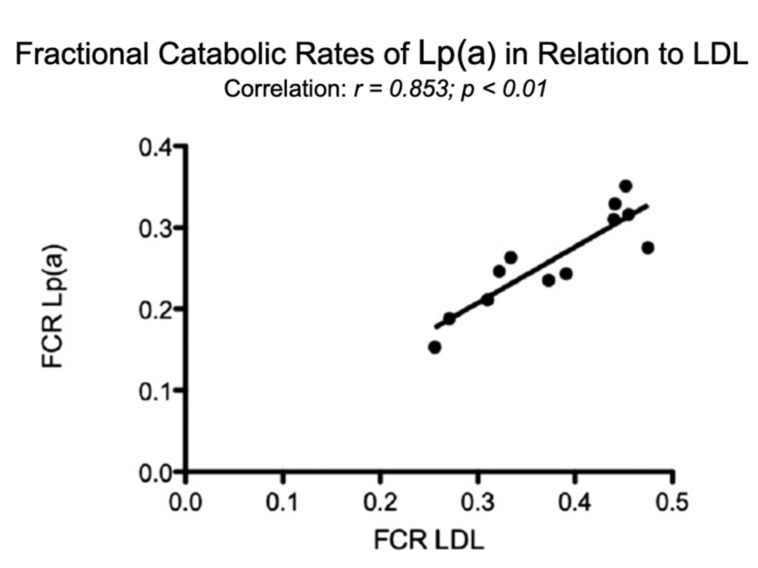
Fractional catabolic rates (FCR) of Lp(a) in relation to the FCR’s of LDL. Turnover studies, as described in Figure 4, were carried out in 12 individuals with various LDL concentrations and the FCR’s of Lp(a), and LDL were analyzed [1].

**Figure 6 ijms-23-03584-f006:**
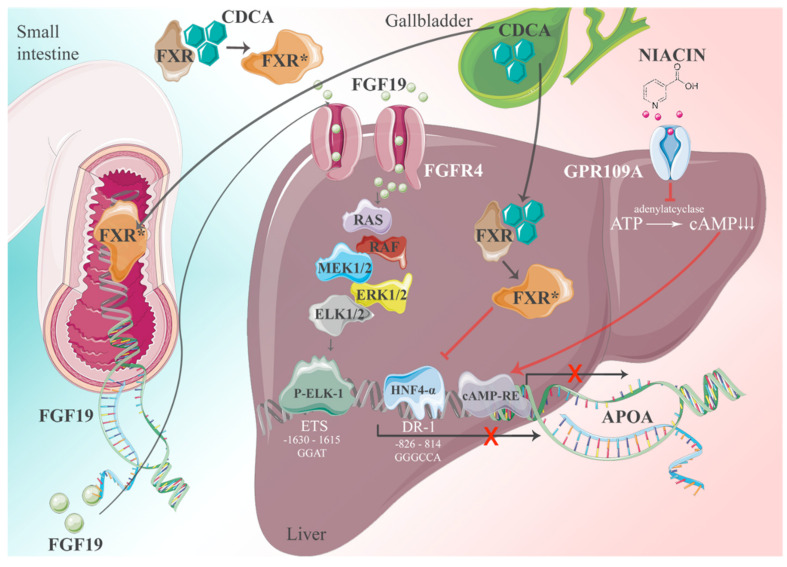
The interference of apo(a) transcription by FXR ligands. The apo(a) transcription is driven by numerous transcription factors. The most important one is probably HNF4α which binds to its response element at −826 to −814 relative to the transcription initiation site. There are two pathways whereby FXR-ligands downregulate the apo(a) expression, one direct one and the other by FGF-19 binding to the FGF4 receptor. There is also a cAMP response element found in the apo(a) promoter that might be inhibited by nicotinic acid.The * indicates activated FXR [1].

**Figure 7 ijms-23-03584-f007:**
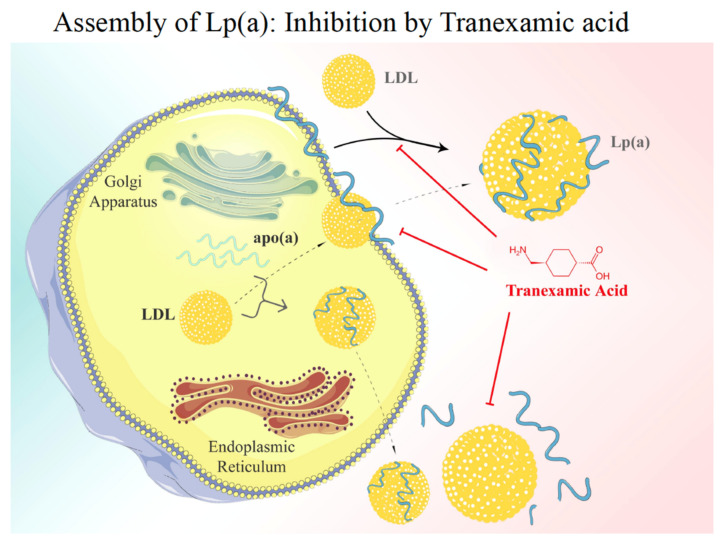
Lp(a) assembly. There are currently two models of Lp(a) assembly discussed: (i) apo(a) is bio-synthesized in the liver and after passage through the Golgi apparatus it binds to the surface of liver cells. Bypassing LDL then associates with apo(a) and, after linking via a disulfide bridge, the native Lp(a) is formed. The first step of assembly may be competed for by Lys analogues such as Tranexamic acid. (ii) Alternatively, the assembly of Lp(a) may take place in the liver cell [1].

**Figure 8 ijms-23-03584-f008:**
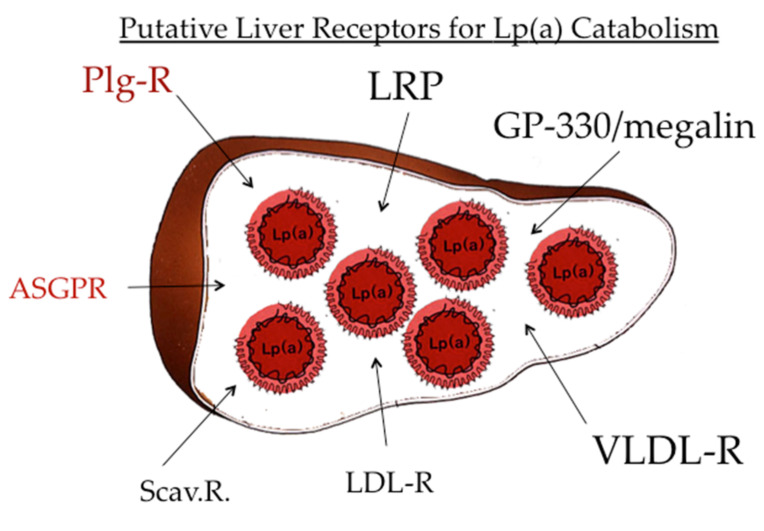
Catabolism of Lp(a). The majority of Lp(a) is not only biosynthesized but also catabolized in the liver. Numerous receptors bind Lp(a) in vitro; however, their impact on in vivo catabolism is not fully explored. Of interest is the binding of Lp(a) to the plasminogen receptor and the asialoglycoprotein receptor.

**Figure 9 ijms-23-03584-f009:**
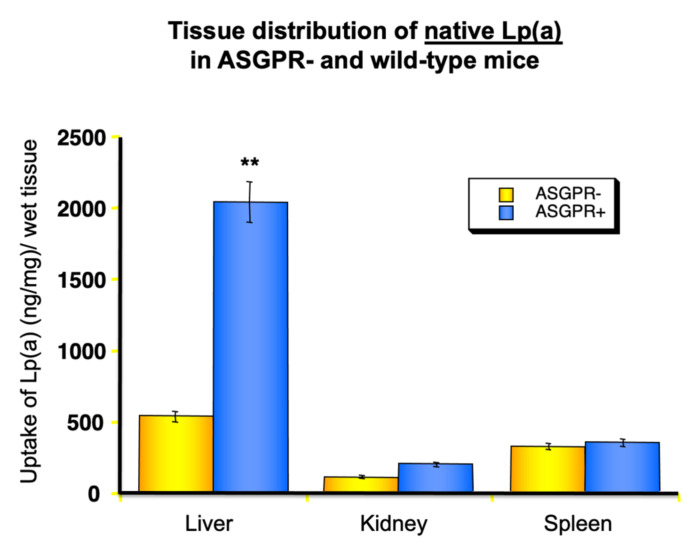
Impact of the asialoglycoprotein receptor (ASGPR) on the uptake of Lp(a) into different organs. Lp(a) was treated with neuraminidase to transform it into asialo-Lp(a), labelled with I^125^ and injected into wild-type mice or transgenic mice lacking the ASGPR (ASGPR-). After 4 h, mice were sacrificed, and the radioactivity found in different organs was measured. Asialo-Lp(a) is primarily taken up by the liver. In addition, Lp(a) from human serum incubated for 24 h at 37 °C in the absence of neuraminidase showed an increased uptake into the liver in comparison to fresh Lp(a) (experiments not shown) [24]. ** *p* < 0.01.

**Figure 10 ijms-23-03584-f010:**
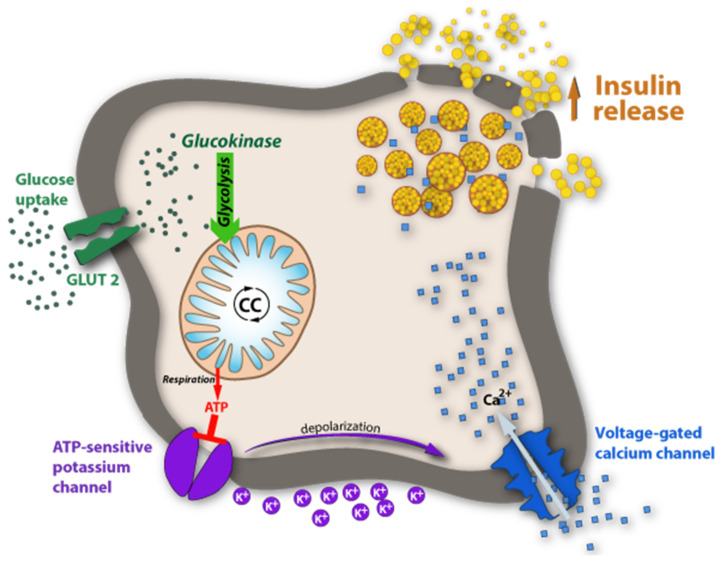
Glucose-sensing by glucokinase. The insulin release is stimulated by high plasma glucose concentrations triggered by a signaling pathway in pancreatic islets through ATP biosynthesis during glycolysis. For details see text.

**Figure 11 ijms-23-03584-f011:**
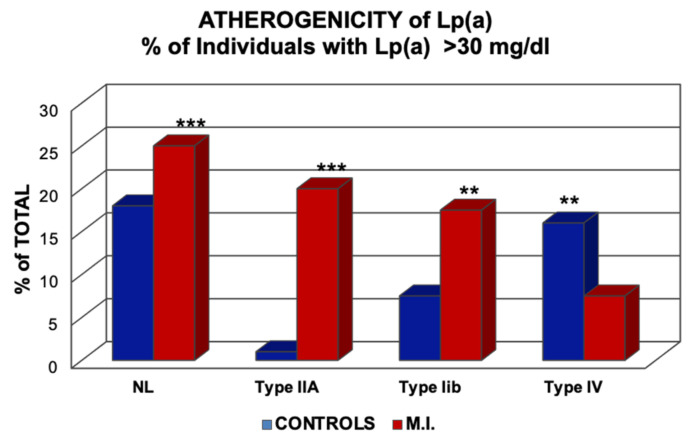
Risk of myocardial infarction (MI) concerning elevated Lp(a). The columns indicate the percentage of individuals with >30 mg/dL Lp(a) in normo-lipemics and patients with hyperlipoproteinemia Types-IIa, IIb, and -IV, according to the Fredrickson classification. ** *p* > 0.01; *** *p* < 0.001. Values shown in the Figure are from Ref. [13].

**Table 1 ijms-23-03584-t001:** Chemical composition of Lp(a) and LDL.

Compound	Lp(a) % *w*/*w*	LDL % *w*/*w*
Protein	30	21
Carbohydrates	10	1.3
Cholsteryl Erster	31.5	42
Free Cholesterol	7	9
Phospholipids	16	20.7
Triglycerides	5.5	6
TOTAL	100	100

The numbers are average values from literature.

## Data Availability

The source of all data reported in this review are cited in the text.

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
