# Peer review of "Lp(a) and the Risk for Cardiovascular Disease: Focus on the Lp(a) Paradox in Diabetes Mellitus"

_ijms, 2022, doi:10.3390/ijms23073584_

Round 1
Reviewer 1 Report
The topic of the review is of first importance. However, the article has many flaws and a lot of information have been provided by the Authors in their previous review (Kostner, K.M.; Kostner, G.M. Lipoprotein (a): a historical appraisal. J Lipid Res 2017, 58, 1-14, doi:10.1194/jlr.R071571). I recommend to rewrite carefully the manuscript, focusing on the topic (a paradoxon) and removing the information provided before. Additional comments can be found below:
- The abstract should be rewritten to be more clearer for the reader. Among others, all abbreviations in the abstract should be explained independently of the main manuscript. English should be corrected. The abstract is too long (the number of words exceeds the journal requirements).
- English should be corrected by a professional in whole manuscript.
- All abbreviations should be explained while first mentioned. The abbreviations used in tables/figures should be explained in the legends independently of the main manuscript.
- In the figures, the results of the authors' own experiments are shown, but the article is a review, not an original article, so it is unconvincing. If the Authors want to present their data, they should prepare an original article containing all required details to assess the quality of the experiments. The ethical consents should also be provided in that occasion.
- In other figures and tables, the source of data/graphs/photos etc. is not provided.
- The different figures are of very different quality (e.g., fig. 3 and 6). It should be unified somehow.
- I find it inappropriate for the Authors to advertise their book in the article (lines 68-71).
- The citation of websites is incorrect (e.g., line 87, line 136).
- A lot of information are not supported by citations.
- The chapter: Lp(a) and diabetes mellitus (DM) - I am not convinced that the detailed description of the insulin release from beta islet cells is required in the review about Lp(a).
- The topic from the title: The Lp(a) Paradoxon in Diabetes Mellitus is described in small part of the article. So, the title does not reflect the content of the review. A lot of information in this review have been previously presented by the Authors in their other works (Kostner, K.M.; Kostner, G.M. Lipoprotein (a): a historical appraisal. J Lipid Res 2017, 58, 1-14, doi:10.1194/jlr.R071571).
- Some figures/graphs have been previously presented by the Authors (Kostner, K.M.; Kostner, G.M. Lipoprotein (a): a historical appraisal. J Lipid Res 2017, 58, 1-14, doi:10.1194/jlr.R071571), so it is a plagiarism for me.
Author Response
Please find enclosed the revised version of our manuscript ijms-1612027. Let me just make a few general statements before dealing with the points in detail that have been criticized.
Although the previous title of our review, “The Lp(a) paradoxon in diabetes mellitus” was rather specific we considered it appropriate to introduce Lp(a) quite extensively as we looked into PubMed for potential papers on Lp(a) in IJMS and found actually zero!
One reviewer also criticized that a lot of information are not supported by citations. Looking into PubMed there are more than 8000 articles listed on Lp(a) and one could extend a review article on this topic by thousands of citations – yet it is not always apparent who was first to mention something in scientific journals or at oral presentations in conferences. I therefore tried to keep everything in balance particularly since any interesting reader nowadays has anyway access to original scientific material through numerous data bases.
Point by point response to Reviewer-1:
- The abstract should be rewritten to be more clearer for the reader. Among others, all abbreviations in the abstract should be explained independently of the main manuscript. English should be corrected. The abstract is too long (the number of words exceeds the journal requirements). The Abstract has been revised as proposed.
- English should be corrected by a professional in whole manuscript. A professional vent over the manuscript and the English language has been corrected.
- All abbreviations should be explained while first mentioned. The abbreviations used in tables/figures should be explained in the legends independently of the main manuscript. Abbreviation are now explained.
- In the figures, the results of the authors' own experiments are shown, but the article is a review, not an original article, so it is unconvincing. If the Authors want to present their data, they should prepare an original article containing all required details to assess the quality of the experiments. The ethical consents should also be provided in that occasion. The figures actually contained only material that had been published in earlier in peer reviewed journals. This is now clearly indicated in the manuscript.
- In other figures and tables, the source of data/graphs/photos etc. is not provided. The graphs and photos had been prepared in the laboratory of G. Kostner and we also hold the copy rights of them. We consider it therefor appropriate to show them in a Review paper – even if they had been already published in previous articles from the same authors.
- The different figures are of very different quality (e.g., fig. 3 and 6). It should be unified somehow. We tried to unify the quality of the figures. Yet the apparent quality depends greatly on the dissolution of the screen! By printing out the manuscript the it becomes apparent that the quality of the Figs is quite OK!
- I find it inappropriate for the Authors to advertise their book in the article (lines 68-71). It was actually not meant as an advertisement – but since the book has not been published yet we removed this citation.
- The citation of websites is incorrect (e.g., line 87, line 136). The citations of the web sites are correct – yet they open only by copy and paste, not by clicking on.
- A lot of information are not supported by citations. Numerous citations have been added. See statement above on top of this reply.
- The chapter: Lp(a) and diabetes mellitus (DM) - I am not convinced that the detailed description of the insulin release from beta islet cells is required in the review about Lp(a). We rather are in favor of keeping this figure since it demonstrates the complexity of the pathway involved in diabetes mellitus – even if it is a simplified cartoon. This should alert the reader that there might be numerous check-points where Lp(a) could interfere with diabetes mellitus and vice-versa that have not been explored yet.
- The topic from the title: The Lp(a) Paradoxon in Diabetes Mellitus is described in small part of the article. So, the title does not reflect the content of the review. A lot of information in this review have been previously presented by the Authors in their other works (Kostner, K.M.; Kostner, G.M. Lipoprotein (a): a historical appraisal. J Lipid Res 2017, 58, 1-14, doi:10.1194/jlr.R071571). We have changed the title and also indicated that part of the information had been published previously. But this is in fact the nature of a review that it reports on published work and is not an original research.
- Some figures/graphs have been previously presented by the Authors (Kostner, K.M.; Kostner, G.M. Lipoprotein (a): a historical appraisal. J Lipid Res 2017, 58, 1-14, doi:10.1194/jlr.R071571), so it is a plagiarism for me. We can hardly follow the considerations of the Reviewer. In order to present a complex pathway impressively and for better understanding we consider such cartoons quite important. Since the figures had been prepared in our laboratory and we hold the copy rights of them – we do not see any plagiarism. What difference would it make if the figures would have been redrawn by us or presented in other styles?
Reviewer 2 Report
For the authors,
It is a good manuscript at least due to two reasons. First, because in the same time it is a well documented literature review of recent data and a description of the author's research group experience.
Secondly, because it is an important medical subject, high Lp(a) being a very high risk of cardiovascular events with or without diabetes mellitus.
Author Response
Thank you for your comments
Round 2
Reviewer 1 Report
The manuscript has been significantly corrected, but I can see some problems still existing, as mentioned below:
- I think that the copyright issue should be explained by the editors of the journal.
- The previous title seemed to be more intringuing. Maybe you could use a combination of the titles: Lp(a) and the Risk for Cardiovascular Disease: Focus on the Lp(a) Paradoxon in Diabetes Mellitus.
- I am not convinced an English professional corrected the manuscript, although it sounds better now. Some language mistakes and numerous typos are still present and should be corrected.
- The abstract should be a total of about 200 words maximum. After the revision, the abstract has 417 words.
- The proper citation of the website should be like mentioned below:
Websites:
9. Title of Site. Available online: URL (accessed on Day Month Year).
Unlike published works, websites may change over time or disappear, so we encourage you create an archive of the cited website using a service such as WebCite. Archived websites should be cited using the link provided as follows:
10. Title of Site. URL (archived on Day Month Year).
Author Response
The manuscript has been significantly corrected, but I can see some problems still existing, as mentioned below:
- I think that the copyright issue should be explained by the editors of the journal. I fully agree
- The previous title seemed to be more intringuing. Maybe you could use a combination of the titles: Lp(a) and the Risk for Cardiovascular Disease: Focus on the Lp(a) Paradoxon in Diabetes Mellitus. The title has been changed as proposed
- I am not convinced an English professional corrected the manuscript, although it sounds better now. Some language mistakes and numerous typos are still present and should be corrected. The whole manuscript has now been thoroughly corrected by an Englishman.
- The abstract should be a total of about 200 words maximum. After the revision, the abstract has 417 words. The abstract has now 200 words.
- The proper citation of the website should be like mentioned below: Sorry- I am not familiar with this type of website citations. Since there were only 2 websites mentioned we removed both of them from manuscript.
Websites:
- Title of Site. Available online: URL (accessed on Day Month Year).
Unlike published works, websites may change over time or disappear, so we encourage you create an archive of the cited website using a service such as WebCite. Archived websites should be cited using the link provided as follows:
10. Title of Site. URL (archived on Day Month Year).